# Does Improvement towards a Normal Cervical Sagittal Configuration Aid in the Management of Lumbosacral Radiculopathy: A Randomized Controlled Trial

**DOI:** 10.3390/jcm11195768

**Published:** 2022-09-29

**Authors:** Ibrahim Moustafa Moustafa, Aliaa Attiah Mohamed Diab, Deed Eric Harrison

**Affiliations:** 1Department of Physiotherapy, College of Health Sciences, University of Sharjah, Sharjah P.O. Box 27272, United Arab Emirates; 2Faculty of Physical Therapy, Cairo University, Giza 12511, Egypt; 3Private Practice and CBP Non-Profit, Inc., Eagle, ID 83616, USA

**Keywords:** randomized controlled trial, traction, disc herniation, cervical lordosis, lumbosacral radiculopathy

## Abstract

A randomized controlled study with a six-month follow-up was conducted to investigate the effects of sagittal head posture correction on 3D spinal posture parameters, back and leg pain, disability, and S1 nerve root function in patients with chronic discogenic lumbosacral radiculopathy (CDLR). Participants included 80 (35 female) patients between 40 and 55 years experiencing CDLR with a definite hypolordotic cervical spine and forward head posture (FHP) and were randomly assigned a comparative treatment control group and a study group. Both groups received TENS therapy and hot packs, additionally, the study group received the Denneroll cervical traction orthotic. Interventions were applied at a frequency of 3 x per week for 10 weeks and groups were followed for an additional 6-months. Radiographic measures included cervical lordosis (CL) from C2–C7 and FHP; postural measurements included: lumbar lordosis, thoracic kyphosis, trunk inclination, lateral deviation, trunk imbalance, surface rotation, and pelvic inclination. Leg and back pain scores, Oswestry Disability Index (ODI), and H-reflex latency and amplitude were measured. Statistically significant differences between the groups at 10 weeks were found: for all postural measures, CL (*p* = 0.001), AHT (*p* = 0.002), H-reflex amplitude (*p* = 0.007) and latency (*p* = 0.001). No significant difference for back pain (*p* = 0.2), leg pain (*p* = 0.1) and ODI (*p* = 0.6) at 10 weeks were identified. Only the study group’s improvements were maintained at the 6-month follow up while the control groups values regressed back to baseline. At the 6-month follow-up, it was identified in the study group that improved cervical lordosis and reduction of FHP were found to have a positive impact on 3D posture parameters, leg and back pain scores, ODI, and H-reflex latency and amplitude.

## 1. Introduction

Lumbosacral radiculopathy associated with disk herniation is one of the most common health-related complaints [1]. Radiculopathy of the S1 nerve root is a frequent pathology, strongly associated with delayed recovery, persistent disability, and increased health care utilization and costs [2]. Despite the high prevalence of this condition [3], its conservative treatment has long remained a challenge for the clinician [4], since there is no strong evidence of the effectiveness of most treatments, particularly for long-term management outcomes [5]. 

The challenge clinicians face is merging an understanding of the patient’s local pathology (e.g., disc herniation) as an etiological factor of their lower back pain (LBP) with an understanding of how altered regional and full spine alignment and biomechanics play a role in the patient’s unique condition. The interaction of tissue pathology and spine dysfunction is clearly ellicudated by Murphy’s concept [6], “pathoanatomy and dysfunction often interact to produce clinical symptoms”. In terms of local biomechanical dysfunction, several investigations have identified that altered trunk posture [7,8] and lumbar spine hypolordotic alignment [9,10] are important etiological factors contributing to the development of and with the presence associated LBP.

In the past decade several publications have identified that head and neck alignment plays a role in whole body pain and impairment including LBP and related disorders [11,12]; whether this is a pure mechanical phenomenon remains unclear. Studies have identified that several of the postural upright postural neurophysiological reflexes, are located within the head and neck region [13]. This implies that correction of the altered cervical spine alignment could be required to achieve optimal full spine postural correction, where the rest of the spine orients itself in a top-down fashion [14]. 

Besides the surgical outcomes of adult spine deformity linking head and neck alignment to health quality of life in thoraco-lumbar deformity patients [11,12], relatively few correlational studies were identified linking cervical spine alignment to thoracic spine ailments and full spine alignment to LBP [15,16]. However, considering the effect that abnormal cervical posture (flexion and translation) has on the stress and strain experienced in the entire spinal cord and nerve roots [17,18], it would seem logical that alterations in cervical spine alignment would influence, at least to some extent, pain and radiculopathy in lumbosacral disorders. It would seem that lumbosacral radiculopathy and LBP conservative treatment today is universally lacking investigations seeking to understand the influence of alignment of the cervical spine relative to pain, disability, and other management outcomes [19,20].

Despite the fact that there is some evidence of a link between lower back pain disorders and head/neck posture [11,12,21], there is limited experimental data to support a cause-and-effect relationship and interventional outcomes. Accordingly, the primary hypothesis of this study was that cervical curve restoration and forward head posture reduction will have short- and long-term effects on three-dimensional (3D) spinal posture parameters as well as lumbar radiculopathy management outcomes such as symptoms, disability, and neurophysiological findings [19,20]. 

In the current study we used a cervical traction orthotic device termed the Denneroll™ to help restore normal sagittal spinal configuration based on principles of 3-point bending traction methods [22]. The Denneroll device uses sustained cervical spine extension loading in a prone position in order to create visco-elastic creep-deformation in the connective tissues of the spine leading to more consistent and effective correction of the cervical sagittal alignment. This was the primary reason for choosing this device to test our working hypothesis.

## 2. Methods

A prospective, randomized, controlled study was conducted at a research laboratory of our university. All the patients were conveniently selected from our institution’s outpatient clinic. Recruitment began after approval was obtained from the Ethics Committee of the Faculty of Physical Therapy, Cairo University; all participants signed informed consent prior to data collection. Patients were recruited from May 2011 to June 2011 for a 10-week treatment investigation with a six-month follow-up. This trial was retrospectively registered at ClinicalTrials.gov (accessed on 10 September 2022) with registration number: NCT05553002.

▪Participant inclusion

*Pain, disability, and symptoms*: Patients were included if they had a confirmed chronic unilateral lumbosacral radiculopathy associated with L5-S1 lumbar disc prolapse with symptoms lasting longer than 3 months to avoid the acute stage of inflammation. All the patients had unilateral leg pain with mild to moderate disability according to the Oswestry Disability Index (ODI) (up to 40%) [23]. All patients had side-to-side H-reflex latency differences of more than 1 ms. Further, patients were selected with lumbar hyperlordosis (sway back posture), which is considered a common posture aberration in CLBP patients [24]. 

▪Participant exclusion criteria

Exclusion criteria included previous history of lumbosacral surgery, metabolic system disorder, cancer, cardiac problems, peripheral neuropathy, history of upper motor neuron lesion, spinal canal stenosis, rheumatoid arthritis, osteoporosis and any lower extremity deformity that might interfere with global postural alignment. The demographic characteristics of the patients are shown in Table 1.

*Radiography*: Participants were screened prior to inclusion by measuring their lateral cervical for a cervical absolute rotatory angle (ARA) formed by two lines intersecting from the posterior body margins of C2–C7 and forward head distance (AHT measured as the horizontal displacement of the posterior superior body corner of C2 vertebra relative to a vertical line extending superiorly from the posterior inferior body corner of C7). Lateral cervical X-rays were obtained with the participant in an upright, neutral, standing posture. If the ARA angle was less than 25° and greater than 0, then a participant was included in the study; thus, straightened and kyphotic cervical curvatures were excluded. Also, if the AHT distance was greater than 15 mm then a participant was included in the study. These X-ray cut-points for ARA and AHT were based on the mean values reported in the study by Harrison et al. [25].

▪Randomization assignment

The patients were randomly assigned to to either the treatment group (*n* = 40) or the control group (*n* = 40). An independent person, blinded to the research protocol and not otherwise involved in the trial, operated the random assignment through picking one of the sealed envelopes which contained numbers chosen by a random number generator. A diagram of patients’ retention and randomization throughout the study is shown in Figure 1.

▪Interventions

The patients in both groups received hot packs (15 min) and TENS therapy to control pain and eliminate the causal role of muscle spasms and/or tightness in changing the posture parameters; these procedures were applied with the patient in the prone position on an examination bench. The TENS treatment was introduced using an electrotherapy device (Phyaction 787, The Netherlands). The TENS therapy was delivered at the lumbosacral region for 20 min. The frequency was set to 80 Hz and pulse width to 50 µs due to its analgesic effect [26]. These conventional treatments were repeated three times per week over the course of 10 weeks for 30 total sessions. Those in the control group received this conventional treatment only. 

The experimental group additionally received Deneroll cervical extension traction (Denneroll Industries (www.denneroll.com (accessed on 10 September 2022)) of Sydney, Australia). Here, the patient lies flat on their back (supine) on the ground with their legs extended and arms by their sides. The patient is encouraged to relax whilst lying on the Denneroll [22]. The Denneroll was placed on the ground and postioned in the posterior aspect of the neck depending on the area needing to be addressed (Figure 2). Participants were screened and tested for tolerance to the extended position on the device to insure they were capable of performing this position; while the Denneroll takes the segments of the cervical spine near the apex of the curve to their end range of extension motion, it does not create hyperextension of the skull relative to the torso. See Figure 2. Patients began with 3 min per session and progressed to a maximum of 20 min per session in an incremental fashion.

The apex of the Denneroll orthotic was placed in one of three regions based on lateral cervical radiographic displacements:In the upper cervical region (C2–C4) region. This position allows extension bending of the upper cervical segments while causing slight anterior head translation (AHT). One subject received this placement location.The apex of the Denneroll orthotic is placed in the mid-cervical region (C4-C6) region. This position allows extension bending of the mid-upper cervical segments while creating a slight posterior head translation. Ten subjects received this placement location.The apex of the Denneroll orthotic is placed in the upper thoracic lower-cervical region (C6-T1) region. This position allows extension bending of the lower to middle cervical segments while creating a significant posterior head translation. Twenty-nine subjects received this placement location.
▪Outcome Measures

A repeatable and reliable method [27,28] was used to quantify the main outcome measurement represented in cervical lordosis (ARA C2–C7) and any amount of anterior head posture AHT (C2–C7). Standard lateral cervical radiographs were obtained at three intervals (pretreatment, 10-weeks post-treatment, and at the six-month follow-up). A representative example of a lateral cervical X-ray in a study group patient at three intervals of measurement is given graphically in Figure 3.

Other outcome measures used to compare effectiveness of the treatment between the study and control groups included the 3D spinal posture parameters, disability, and neurophysiological findings.

Rasterstereography (Formetric 2, Diers International GmbH, Schlangenbad, Germany) was used to examine posture and back shape characteristics. All testing procedures were done following Lippold et al.’s protocol [29]. The Formetric scans were taken in a relaxed standing position. The patient was positioned in front of the black background screen at a distance of two meters from the measurement system. The column height was aligned to move the relevant parts of the patient’s back into the center of the control monitor by using the column up/down button of the control unit; to ensure the best lateral and longitudinal position of the patient a permanent mark on the floor was used. The patient’s back surface (including upper buttocks) was completely bare in order to avoid image disturbing structures.

After the patient and the system were correctly positioned, the system was ready for image recording. The image processing consisted of automatic back surface reconstruction and shape analysis. The sagittale plane parameters (lumbar angles, thoracic angles, and trunk inclination), the frontal plane parameters (trunk imbalance and lateral deviation) and the transversal plane parameters (vertebral surface rotation and pelvis torsion) were selected to cover the posture profile in three planes. A representative example of the Formetric system’s print out is given graphically for a study group participant (Figure 4).

Disability was measured using the Oswestry Disability Index. The total score is transferred onto a scale ranging from 0 to 100, where 0 indicates no disability and 100 indicates worst possible disability [23].

The back and leg pain intensity were measured using the numerical pain rating scale (NPRS), which is considered a valid and reliable scale [30]. The patients were asked to place a mark along a line to denote their pain level; 0 reflecting ‘‘no pain’’ and 10 reflecting the ‘‘worst pain’’.

Latency and peak-to-peak amplitude of the H-reflex, the recommended H-reflex diagnostic criteria for lumbosacral radiculopathy [20], were used in the current study. An electromyogram device (Tonneisneuroscreen plus version 1.59, Germany) was used to measure this variable for all patients before starting the treatment, at the end of 10 weeks, and at the six-month follow-up. All testing procedures were done following Al-Abdulwahab and Al-Jabrb’s protocol [31]. The patient was lying supine on a wooden padded table, with arms on the side. The knee was flexed 20 degrees by placing a small cushion under the knee to relax the gastrocnemius muscle. The tibial nerve was stimulated at the popliteal fossa, midway between the tendon of the biceps femoris and semimembranosus, using a silver–silver chloride surface-stimulating bar electrode with the cathode proximal to the anode.

For recording, surface bar electrodes were placed 3 cm distal to the bifurcation of the gastrocnemii and superior to the Achilles tendon. A ground surface metal electrode was positioned midway between the stimulation and recording electrodes to minimize the stimulus artifact. Before attaching the recording electrodes, the underlying skin was shaved and cleaned with a piece of cotton soaked with alcohol. The stimulation parameters were 1.0 ms pulse duration and intensity that elicited H-maximum with minimum and stable M-response at a frequency of 0.2 Hz. Four readings of the maximum H-reflex with minimum and stable M-responses were recorded and averaged from the involved leg. The signals were amplified 500–2000 *×* using differential amplification and filtered at 3 Hz–10 kHz, digitized (10 kHz) and stored on a computer for analysis.

▪Sample size determination

A prior power calculation indicated that 27 patients were needed in each group to detect a difference in cervical lordotic angle between the groups with 90% power and a 5% significance level; a 2-tailed test, and an expected effect size of *d* = 0.9 based on a pilot study consisted of 10 patients who received the same program. The sample size estimation was based on an unpublished pilot randomized controlled clinical trial that used a similar protocol for patients with discogenic lumbosacral radiculopathy. The population was in the same age range with minimal change in the control treatment (stretching vs. hot backs herein). In this pilot, the traditional therapy was TENS, back and lower limb stretching exercises, and ARA C2–C7 cervical lordosis for our primary outcome. The pilot project had no long-term follow-up; therefore, the sample size was calculated based on pre-post lordosis changes. The mean change and standard deviation of the cervical lordosis were estimated at 3.2 and 3.7, respectively. To account for the possibility of significant drop-out rates, the sample size was increased by 40%.

▪Data analysis

To compare the experimental group and the control group, statistical analysis was based on the intention-to-treat principle, and *p*-values less than 0.05 were considered significant. We used multiple imputations to handle missing data. To impute the missing data, we constructed multiple regression models including variables potentially related to the fact that the data were missing and also variables correlated with that outcome. We used Stata (Stata Corp, College Station, TX, USA). The 2-way repeated-measures analysis of covariance was used to compare between groups. The model included one independent factor (group), one repeated measure (time), and an interaction factor (group * time). The baseline values of the outcomes were used as covariates to assess the between-group differences, to center the baseline covariates, everyone’s score value was subtracted from the overall mean. A *t*-test at two follow-up points (after 10 weeks of treatment and at the six-month follow-up) was performed to test the between group differences at the different intervals.

## 3. Results

A diagram of patients’ retention and randomization throughout the study is shown in Figure 1. One hundred and fifty patients were initially screened. After the screening process, 84 patients were eligible to participate in the study and 80 completed the first follow up at 10 weeks, while 71 of them completed the entire study including the 6-month follow up. The study design did not include a pre-determined adverse event protocol. However, participants were formally asked during their treatment sessions if they were experiencing any unusual adverse events or increased pain due to the interventions. No adverse events were documented by the treating therapist aside from minimal and transient discomfort in the neck as the patient acclimatized to using the Denneroll device at the point of cervical spine contact.

Results are summarized and presented as mean (±SD) in Table 2. After 10 weeks of treatment, the analysis of covariance (ANCOVA) revealed a significant difference between the study and control groups adjusted to baseline values for all following variables: ARA (*p* = 0.001), AHT (*p* = 0.002), neurophysiological findings represented in H-reflex amplitude (*p* = 0.007) and H-reflex latency (*p* = 0.001); 3D postural parameters in terms of trunk inclination (*p* = 0.001), lumbar lordosis (*p* = 0.002), thoracic kyphosis (*p* = 0.001), trunk imbalance (*p* = 0.001), pelvic inclination (*p* = 0.005), and surface rotation (*p* = 0.01). 

At the six-month follow-up, the analysis showed that there were still significant differences between the study and control groups for all the previous variables: radiographic measurements of cervical lordosis ARA (*p* = 0.01), AHT (*p*= 0.028); neurophysiological findings represented in H-reflex amplitude (*p* < 0.001) and H-reflex latency (*p* = 0.004); as well as the 3D postural parameters of trunk inclination (*p* = 0.04), lumbar lordosis (*p* = 0.001), thoracic kyphosis (*p* = 0.001), trunk imbalance (*p* < 0.001), pelvic inclination (*p* = 0.02), and surface rotation (*p* = 0.05). Table 2 presents this data. Figure 3 depicts an example of radiographic changes in the study group across the 3 time periods of evaluation. Figure 4 depicts an example of the 3D posture changes in the study group across the 3 time periods of evaluation.

At the 10-week post-treatment analysis, for back pain, leg pain and the ODI disability index, the between group analysis revealed insignificant difference between the groups at the first measurement interval: back pain, *p* = 0.27; leg pain, *p* = 0.1; and ODI, *p* = 0.6. In contrast, at the 6-month follow up, there was statistically significant differences between the groups for back pain (*p* < 0.001), leg pain (*p* < 0.001), and ODI (*p* < 0.001). These data are reported in Table 2. Specifically, the 6-month follow up data indicated that the control group’s scores regressed back to pre-intervention levels while the study groups’ improvements in these variables were maintained.

## 4. Discussion

This study tested the hypothesis that correction of sagittal cervical alignment would influence management outcomes of chronic lumbosacral radiculopathy. We compared TENS and hot packs in a control group to the outcomes of a study group receiving the control interventions plus the addition of an extension cervical traction device (the Denneroll) known to correct sagittal cervical spine alignment. [22] As expected, after 10 weeks of treatment, the study group (traction group) was found to have improvements in the cervical lordosis and anterior head translation compared to no change in the control group. Additionally, at 10 weeks, the study group was found to have improved 3D thoraco-lumbar-pelvic posture as well as improved neurophysiology as measured with the H-reflex. 

Unexpectedly, the patient perceptive outcomes of lower back pain, leg pain, and lower back disability showed no differences between the groups; both groups improved equally at 10 weeks of treatment. However, after the 6-month follow-up with no further interventions, the control groups improvements regressed back to baseline values while the study group showed improved lumbar radiculopathy management outcomes for all variables. Thus, the difference in our study groups 6-month outcomes compared to the control group of improved radiographic, 3D postural parameters, clinical, and neurophysiological variables all indicate that our hypothesis is supported; improved cervical sagittal alignment does have a significant effect on the management outcomes of lumbosacral radiculopathy.

▪Back pain, leg pain improvements at 10 weeks

The outcomes of back pain, leg pain, and disability for both the study group and control group showed similar improvements at the 10-week post-treatment assessment. The temporal reduction of pain in both our groups can be attributed to the short-term effect of TENS and hotpacks. For instance, Escortell-Mayor et al. [32] reported that the effect of TENS significantly decreased 6 months after the intervention. Similarly, the systematic review of Gaid and Cozens [33] provides evidence to support the use of TENS as a short-term effective treatment modality for chronic lower back pain. This is likely the explanation for the worsening (waning of treatment effect) of pain and disability in our control group at 6 months.

▪Sagittal Cervical Alignment

The improvement in the forward head posture and cervical lordotic curve recorded by the study group receiving the Denneroll was anticipated in as much as previous investigations have identified that this device does indeed improve the cervical lordosis and reduce anterior head translation [22]. Sustained extension loading on devices like the Denneroll causes stretching of the visco-elastic tissues (discs, ligaments, muscles) of the cervical spine in the direction of the neutral head and neck posture and increased lordosis; this is the likely explanation and rationale for sustained extension loading restoring the cervical lordosis and improving anterior head translation [22,34,35]. 

Our study identified a smaller mean improvement in cervical lordosis compared to previous investigations using extension traction devices [22,34,35]. These smaller mean changes are likely a result of our use of only 30 sessions on the Denneroll and the pre-determined inclusion criteria of hypolordosis with AHT, thoracic hyperkyphosis, and lumbar hyperlordosis for subject participation. It is possible that if we allowed straight and reversed cervical curves in our population, the corrections would have been greater as the potential for improvement would be more. We suggest it is likely that the improved cervical sagittal alignment played a role in the improved outcomes of lumbosacral radiculopathy in our study group for the reasons discussed below.

▪3-D Posture Changes

The study group receiving the Denneroll traction experienced significant changes in posture parameters occurring in the sagittal, transverse, and coronal planes. These postural changes suggest an important role for the cervical spine on global spinal posture via complex neurophysiological reflex mechanisms [13]. For instance, studies have identified neurological regulation of static upright human posture that is largely dependent on head posture [36,37] and consequently a normal joint afferentation process. 

Our results are conceptually in agreement with Lewit [31] who highlighted the association between head posture and the pelvo-ocular reflex, where an anterior pelvic translation to balance the head’s center of gravity may occur; this interdependence between body segments has been reported by others as well [38,39]. Additionally, in the study group receiving the Denneroll traction, the resultant changes in the sagittal contour of the whole spine may have contributed to the significant improvements of posture parameters in the transverse and coronal planes as well. For example, the relationship between the sagittal and coronal spinal contours [40,41,42] and between the sagittal configuration of the spine and axial rotation displacements [43] has been detailed.

It is likely that the continuous asymmetrical loading from altered postures (forward head posture, loss of cervical lordosis, sagittal, transverse and coronal displacements) may be the possible explanation for the decline in the functional status for the control group at the 6-month follow up as supported by predictions from experimental and biomechanical spine-posture modelling studies [44,45], surgical outcomes [11,12] and large cohort investigations [15]. Abnormal posture is considered as a predisposing factor for pain because it elicits abnormal stresses and strains in many structures, including bones, intervertebral discs, facet joints, musculotendinous tissues, and neural elements [11,12,17,18,44,45]. Thus, the 6-month improvement of pain for the study group seems reasonably attributable to the restoration of normal posture.

In contrast to our findings, other studies in the literature have reported that postural abnormalities were of minor importance for LBP and disability [46,47,48]. The lack of a clear correlation between sagittal spine curves and health was suggested in a systematic review conducted by Christensen and Hartvigsen [47]. However, the contradictory findings between the correlation between posture and pain in previous studies might simply be due to a lack of uniform classification and measures; most of the previous research is based on 2D posture analysis and poor experimental design. Further, when taken as a whole, comprehensive literature reviews including systematic literature reviews and meta-analyses on the topic, suggests a correlation between sagittal plane posture and patient outcomes [11,12,15,49,50]; especially in the cervical spine [51,52,53].

▪Neurophysiological improvements

The current investigation assessed neurophysiological responses at the nerve root by evaluating the H-reflex. Notably, we identified significantly improved H-reflex latency and amplitude in the study group compared with the control group at the 10-week evaluation and this improvement was maintained at the 6-month follow-up. The only explanation that seems reasonable herein, is that improved posture and cervical spine alignment in the study group reduced longitudinal stress and strain in the central nervous system and in the lumbosacral nerve roots. This concept is supported by biomechanical investigations confirming that abnormal posture of any part of the spinal column will induce abnormal stresses in the entire cord and nerve roots while normal posture will minimize these stresses [17,18]. This concept of altered postures of the thoraco-lumbar spine increasing tension on the nerve root and increasing the likelihood of radiculopathy has been documented elsewhere [54].

Specific to the cervical spine, Breig and Marions [18] demonstrated the effect that slight cervical spine flexion (straightening of the cervical lordosis) has on the lumbosacral nerve roots where increased tension was found as far down as the cauda equina and the sacral plexus. With loss of the cervical lordosis causing increased tension in the lumbosacral nerve roots, a disc herniation in the lumbosacral region would be associated with an increased shear load at the interface between the disc and the nerve root [17]. Finally, it has recently been confirmed that improvement of the sagittal cervical radiographic alignment does improve neurophysiological amplitudies and latencies of somatosensory evoked potentials in the cervical spine, both measured in the peripheral (nerve root) and central systems (central condition time) [22,55]. Thus, it seems logical that our study findings indicate that improved cervical sagittal alignment and improved 3D posture were the explanations for the improvements in the H-reflex identified in our study group.

▪Study limitations

Our study has some potential limitations, each of which points towards directions of future investigations. The primary limitations were the lack of investigator blinding and the sample was a convenient sample rather than a random sample of the whole population. Further, it remains to be seen what effect a greater frequency and number of traction sessions will produce and what effect the Denneroll would have on improvement of altered cervical curves with other types of primary lumbar disorders. 

## 5. Conclusions

Our study identified that both groups experienced improvement in lower back pain, leg pain and disability levels after 10 weeks (30 sessions) of interventions. However, cervical lordosis, 3D posture of the trunk and the neurophysiological findings, represented in the H-reflex, identified greater improvements in the study group receiving the Denneroll. At the 6-month follow up, the control groups improvement in lower back pain, leg pain and disability reverted back to pre-study values. In contrast, at the 6-month follow-up the Denneroll traction study group showed improvements in all variables, including lower back pain, leg pain, disability, the 3D posture parameters, neurophysiological, and sagittal cervical alignment. These findings suggest that improving the cervical sagittal radiographic alignment offers benefits to this population suffering from chronic lower back pain and lumbosacral radiculopathy. 

## Figures and Tables

**Figure 1 jcm-11-05768-f001:**
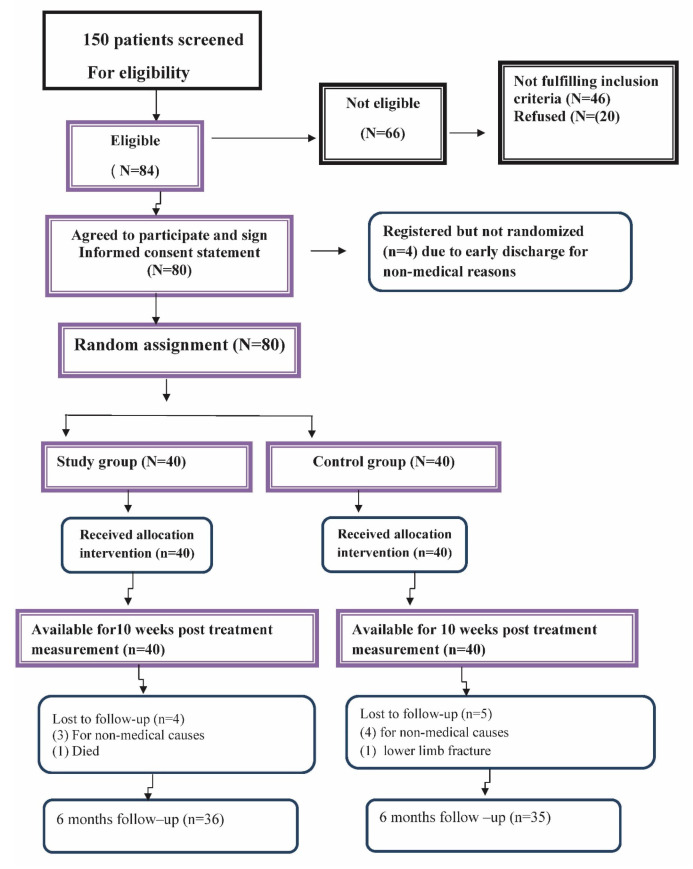
Flow of study participants.

**Figure 2 jcm-11-05768-f002:**
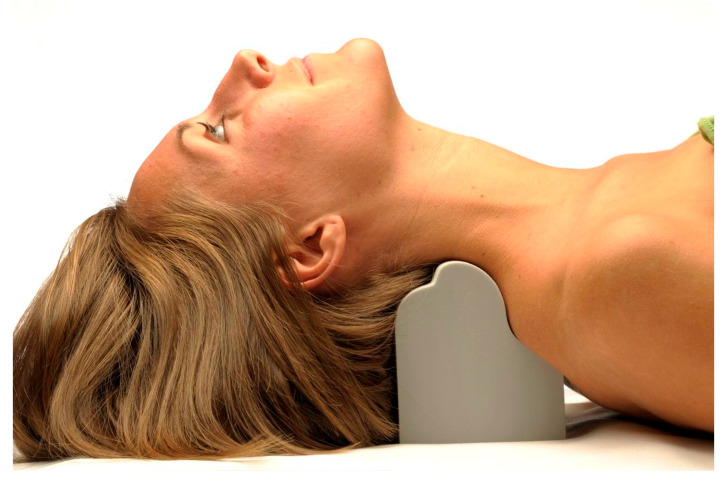
The Denneroll cervical traction orthotic. The participant must lie on a firm surface, such as the floor, and place the peak of the Denneroll just distal to the apex of their cervical lordotic abnormality as shown on the lateral cervical X-ray. Shown is a mid-cervical spine placement. ©Copyright CBP Seminars. Reprinted with permission.

**Figure 3 jcm-11-05768-f003:**
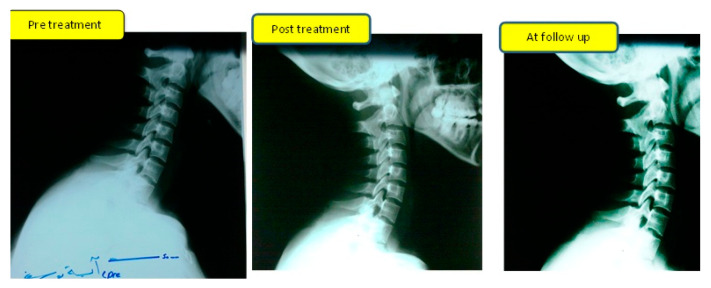
Sample of lateral cervical X-ray findings of a participant in the study group receiving Denneroll traction application at the three intervals of measurement. Pre-treatment prior to study participation, 10-week post-treatment participation, and at the 6-month study follow up radiographs are shown demonstrating improved cervical lordosis and reduced anterior head posture.

**Figure 4 jcm-11-05768-f004:**
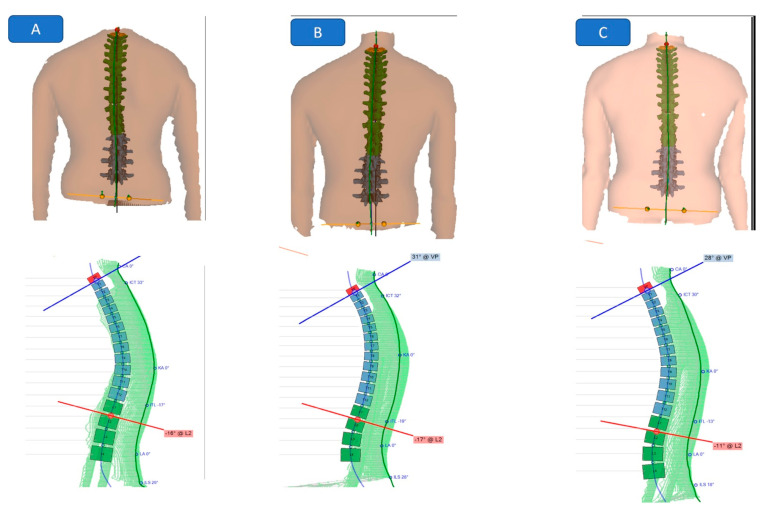
Formetric findings at the three intervals of measurement for a representative patient in the study group: In the left-hand column is the sagittal plane surface profile of the thoracic and lumbar spines while the right-hand column is the posterior view of the coronal and transverse aspects of posture deformity for the thoracic, lumbar, and top of the pelvic regions. (**A**) pre-treatment; (**B**) 10-weeks post-treatment; and (**C**) is the 6-month follow up.

**Table 1 jcm-11-05768-t001:** Baseline participant demographics and tests of significance of between group variables.

	Study Group (*n* = 40)	Control Group (*n* = 40)	*p* ‡
Age (y)	46.3 ± 2.05	45.9 ± 2.1	0.391
Height (cm)	172 ± 9	175 ± 10	0.162
Weight (kg)	75 ± 9	80 ± 10	0.021 *
**Gender**			1.000
Male	22	23	
Female	18	17	
**Work**			0.087
Sedentary	25	18	
Mobile	11	10	
Sedentary and mobile	4	12	
**Previous back pain treatment (yes/no)**			0.580
Surgery	0	0	
Medication	29	30	
Physical therapy	5	7	
Other	6	3	

‡: Two-sided 2-sample t test for continuous variables and Fisher’s exact test for categorical variables. SD: Standard deviation; values are mean (±SD) for age, height, weight and number for the term ‘other’. *: Statistically significant difference between groups for weight.

**Table 2 jcm-11-05768-t002:** Means, standard deviations (±SD), and statistical significance for all outcome variables in the control group versus the study subjects at initial, 10 weeks of treatment, and 6-month follow up.

Dependent Variables		Initial Baseline	10-Weeks Post	6-Month Follow Up	*p*-Value		
					G	T	G*T
**Trunk inclination**	Study G	6 ± 1.0	5.1 ± 1.1	5.5 ± 1.4	<0.001	<0.001	<0.001
	Control G	6.7 ± 1.3	6.5 ± 1.1	6.8 ± 1.3			
	Between group analysis		0.01	0.04			
**Thoracic kyphosis**	Study G	64.9 ± 4.2	62.0 ± 5.3	63.1 ± 5.1	<0.001	<0.001	<0.001
	Control G	62.2 ± 4.9	61.5 ± 4.9	61.9 ± 5.2			
	Between group analysis		0.001	0.001			
**Lumbar lordosis**	Study G	49.5 ± 3.4	46.7 ± 3.5	47.1 ± 3.3	<0.001	<0.001	<0.001
	Control G	49.1 ± 3.2	48.3 ± 3.2	48.9 ± 3.4			
	Between group analysis		0.002	0.001			
**Trunk imbalance**	Study G	20.4 ± 2.9	17.4 ± 2.8	17.8 ± 2.7	<0.001	<0.001	<0.001
	Control G	20.1 ± 2.9	19.3 ± 2.4	19.5 ± 2.6			
	Between group analysis		0.001	<0.001			
**Pelvic inclination**	Study G	3.2 ± 0.6	1.9 ± 0.8	2.0 ± 1	<0.001	<0.001	<0.001
	Control G	3.0 ± 0.6	3.0 ± 0.9	3.3 ± 0.8			
	Between group analysis		0.005	0.02			
**Surface rotation**	Study G	5.6 ± 1.1	5.01 ± 1.3	5.6 ± 1.6	<0.001	<0.001	<0.001
	Control G	6.4 ± 1.0	6.3 ± 0.9	6.7 ± 1.0			
	Between group analysis		0.01	0.05			
**^+^ Cervical ARA**	Study G	13.3 ± 3	18.25 ± 2.6	17.6 ± 2.8	<0.001	<0.001	<0.001
	Control G	13.5 ± 2.7	14 ± 2.8	14 ± 2.9			
	Between group analysis		0.001	0.01			
**Functional index**	Study G	29 ± 5.6	25.3 ± 5.4	25 ± 5	<0.001	<0.001	<0.001
	Control G	31.9 ± 5.8	31.6 ± 5.5	33 ± 6.2			
	Between group analysis		0.6	<0.001			
**H-reflex amplitude**	Study G	2.4 ± 0.3	2.8 ± 0.4	2.7 ± 0.3	<0.001	<0.001	<0.001
	Control G	1.9 ± 0.2	2.1 ± 0.4	2 ± 0.6			
	Between group analysis		0.007	<0.001			
**H-reflex latency**	Study G	33.5 ± 0.7	32.4 ± 0.7	32.5 ± 0.6	<0.001	<0.001	<0.001
	Control G	33.8 ± 0.6	33.5 ± 1.1	34 ± 2.1			
	Between group analysis		0.001	0.004			
**Back pain**	Study G	5.2 ± 0.8	3.5 ± 1.1	3.3 ± 1.5	<0.001	<0.001	<0.001
	Control G	4.6 ± 1	3.4 ± 1	4.7 ± 1.5			
	Between group analysis		0.27	<0.001			
**Leg pain**	Study G	6.9 ± 0.7	4.8 ± 1.3	4.7 ± 1.5	<0.001	<0.001	<0.001
	Control G	6.4 ± 1.1	4.7 ± 1.4	6.1 ± 1.6			
	Between group analysis		0.1	<0.001			
**^++^ AHT**	Study G	26.5 ± 5.7	21 ± 5.3	22.0 ± 5.3	<0.001	<0.001	<0.001
	Control G	26.1 ± 3.9	24.9 ± 3.8	25.3 ± 3.2			
	Between group analysis		0.002	0.028			

T2-way repeated-measures analysis of covariance was used to compare between groups. The model included one independent factor (group: G), one repeated measure (time: T), and an interaction factor (group * time: G*T). ^+^ARA: Absolute rotation angle for cervical lordosis along the backs of vertebral body margins of C2 and C7. ^++^AHT: Forward or anterior head translation posture.

## Data Availability

The datasets analyzed in the current study are available from the corresponding author on reasonable request.

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
