# Peer review of "Does Improvement towards a Normal Cervical Sagittal Configuration Aid in the Management of Lumbosacral Radiculopathy: A Randomized Controlled Trial"

_jcm, 2022, doi:10.3390/jcm11195768_

Round 1

Reviewer 1 Report

This manuscript is of great interest to physical therapists treating cervical spine dysfunction. However, there are aspects that concern me and that must be resolved:

The need for the study is not well justified in the introduction. There is not enough evidence to support the study proposal. It is necessary to justify why this device is going to be positive for patients.

Regarding the registration of the study, it must be registered on ClinicalTrials.gov. It is required for any publication involving a clinical trial regardless of country.

The measurements do not specify the position of the patient in which they were measured.

Were adverse effects recorded? how were they valued?

I do not understand the purpose of applying the device to the neck since it generates an extension, a movement that is symptomatic in many patients. In addition, did that position not generate dizziness or discomfort in the patient?

Regarding the calculation of the sample size, how was the pilot study carried out? In what conditions? in which patients and how was the intervention performed?

The discussion should be reconsidered since I am concerned that it is generalized and it is considered that the improvements in the study are due to certain neurophysiological mechanisms that have not been studied or evidenced.

Be more concise in the findings in your study and give an explanation based on previous evidence.

The authors should also be more cautious in writing the conclusion.

Reviewer 2 Report

Review of manuscript titled “Does improvement towards a normal cervical sagittal configuration aid in the management of lumbosacral radiculopathy: A 3 randomized crontolled trial”. The authors have investigated the effects of the role of cervical sagittal configuration on the global spine posture. The authors found statistically significant changes between the control group and the Denneroll traction could be a valuable non-invasive treatment for patients with lumbosacral radiculopathy. The figures are illustrative. I have some comments.

There seems to be a misspelling in the title and some grammatical errors in the manuscript.

Abstract

The authors state: “Restoring cervical lordosis and reduction of FHP with Denneroll traction was found to have a positive impact on 3D posture paramters, leg and back pain scores, ODI, and H reflex latency and amplitude.”

But it seems like there were to statistical significance for back pain, leg pain and ODI at 10 weeks? Perhaps clarify that there were statistically significant differences at 6 months follow-up for back and leg pain.

Methods

Perhaps the clinical inclusion criteria should be before the radiological? Or were they first identified through their radiological examination?

Under which time period were the patients included? Perhaps move up from Declarations.
